# Seamless Switching Control Technology for the Grid-Connected Converter in Micro-Grids

**Changli Shi** [1,2]**, Tongzhen Wei** [1,2]**, Yushu Sun** [1,2,]*****, Dongqiang Jia** [3] **and Tianchu Li** [4]

[1]   Institute of Electrical Engineering, Chinese Academy of Sciences, Beijing 100190, China;
      shichangli@mail.iee.ac.cn (C.S.); tzwei@mail.iee.ac.cn (T.W.)
[2]   University of Chinese Academy of Sciences, Beijing 100049, China
[3]   Beijing Electric Power Research Institute, State Grid Beijing Electric Power Company, Beijing 100075, China;
      jdq@126.com
[4]   Electric Power Research Institute of Hainan Power Grid Co., Ltd., Haikou 570311, China;
      hellosky77@126.com
*****   Correspondence: yushusun@mail.iee.ac.cn; Tel.: +86-1776-765-7353

**Abstract:** In order to ensure the reliable power supply of the local load in the micro-grid (MG), a seamless switching control technology (SSCT) suitable for grid-connected converter (GCC) is proposed. This technology includes silicon-controlled rectifiers (SCR) forced shutdown control strategy (SCR-FSCS) and three-loop control strategy (TLCS). The SCR-SSCT adjusts the load voltage in real time to form a back voltage at the grid-connected inductor, which greatly reduces the SCR shutdown time and ensures the reliability of local load power supply. The TLCS can easily realize the switching between the current source mode and the voltage source mode of the GCC. An experimental platform is established to carry out the relevant experiments. The experimental results show the rationality and effectiveness of the theoretical analysis and the proposed control technology.

**Keywords:** grid-connected converter; forced shutdown strategy; three-loop control strategy; seamless switching control technology

---

## 1. Introduction

### 1.1. General Context

In order to solve the problem of global climate change, renewable energy sources such as wind energy and photovoltaics have developed rapidly [1,2]. micro-grid (MG) is an effective form of renewable energy utilization has become the consensus of the scientific community [2]. MG is a small power generation and distribution system that connects wind power, photovoltaic and other power generation types, energy storage, and loads through power electronic equipment. MG can not only improve the utilization efficiency of renewable energy but also accurately track load changes to ensure the stable power supply of the load [3,4]. Based on the concept of MG, various countries have carried out demonstration projects. The Consortium for Electric Reliability Technology Solutions (CERTS) of the United States built a demonstration project at the Dolan Technology Center [5]. The goal of this project is to verify the stability of voltage and frequency under islanding conditions, as well as seamless switching between operating modes [5]. There are many MG projects built by the European Union [5]. The main goal of the Labein project in Spain is to verify the performance of the system control strategy under grid-connected conditions. Japan has built the Sendai MG project [6]. The project provided uninterrupted power supply to the loads in its service area during the 2012 Japanese earthquake. This experience is of great significance to the development of MG. The Bulyansungwe MG project in

Uganda is of great significance to the discussion of power supply modes in areas with insufficient power and abundant sunlight resources [5].

### 1.2. Motivation

MG have various working modes [7–10]. They can work in the grid-connected mode (GCM) to exchange energy with the grid and in the off-grid mode (OGM), when grid maintenance is needed or grid faults have occurred, to independently supply power to local loads and ensure the continuity of the load power consumption. This requires the GCC in the MG to ensure the continuity of the load voltage and phase in the process of grid-connection/off-grid switching. That is, it must have a seamless switching function between GCM and OGM. Therefore, it is of great significance to carry out the research on the SSCT.

### 1.3. Brief State of the Art

Scholars at home and abroad have carried out a lot of research on seamless switching control technology for inverters. In [11], an optimal design scheme for grid-connected controllable switches in MG was proposed, but the scheme realized seamless switching by adding passive networks, which increased the difficulty of coordinated control and the cost of the equipment. In [12], the causes of voltage and current impacts in the process of the dual-mode switching of inverters between GCM and OGM were analyzed, and a seamless dual-mode switching method of inverters based on a nonlinear droop curve, which can suppress the impact and distortion of voltage and current during the switching process was proposed. In [13], control models of an electric vehicle charging/discharging/storage integrated power station in the V2G mode, independent mode, and switching mode were established, and a centralized multi-mode unified operation control strategy was proposed to realize seamless switching of operation modes. Based on virtual synchronous generator control technology, a pre-synchronization control strategy for GCM and OGM, which realized seamless switching between different operation modes, was proposed in [14–18]. However, the strategy mainly realized smooth switching between different modes from the perspective of software control, without considering the coordination with the control switch of GCM and OGM. In [19], considering the islanding detection problem during the seamless switching of GCC, an anti-islanding detection method was proposed to improve the adaptability of the islanding detection method, but further research on how to control the inverter after islanding detection remains necessary. In [20], considering the problem that voltage is uncontrollable during islanding detection when the GCM of a master-slave MG is switched to OGM, a voltage and current cooperative control strategy was proposed to ensure the continuity of the control mode. A SSCT based on positive sequence extractor was proposed in [21]. A SSCT based on a virtual synchronous machine was proposed in [22]. The pre-grid-connected mode was introduced. However, during the switching process, the power required by the load was provided by the power grid, which had an adverse effect on the grid. Based on the instantaneous power theory, model predictive control technology was introduced in [23] to improve the dynamic performance of the GCC. In [24], a seamless switching inverter for a MG was designed, and strategies, such as sine wave pre-detection, fast commutation current compensation, and predictive conversion voltage control, were proposed to realize SSCT. In [25], four typical operation modes of an AC/DC hybrid MG with multi-bus structures were designed, and corresponding mode switching strategies and implementation logic were proposed. However, the switching strategy mainly focused on the upper-level energy management of the system, and no specific implementation form of seamless switching of grid-connected equipment was proposed. According to the working characteristics of the energy storage converter in MG [26], a SSCT was proposed. However, the coordination control problem of the software control strategy and hardware switch was not studied in this scheme, which influenced the control effect of the strategy. A master–slave coordinated control strategy for distributed new energy power generation units connected to the common connection point through two parallel

inverters was proposed in [27,28]. However, the strategy mainly focused on the coordinated control of parallel inverters, and the seamless switching process of a single inverter was not analyzed in detail.

In the field of multi-loop strategies for GCC, A multi-loop control technology for GCC was proposed in [29]. It was mainly used to suppress harmonic distortion and frequency variation, and the application of this strategy to a seamless switching control has not been studied [30,31] proposed a multi-loop control framework for voltage source GCC using LCL filters. A sliding mode controller was used in the inner loop to make the inverter become a current source inverter with CL filter. For the voltage source inverter adopting the LCL filter, the damping characteristic of the inverter is improved by adopting multi-loop control, and the stability of the system was enhanced [32]. For the boost inverter, a multi-loop control strategy based on sliding mode control was proposed to improve the output voltage quality of the inverter and reduce the voltage stress of semiconductor switch [33].

It can be seen, from the above analysis, that the existing studies in this field focus on the software control strategy, and there are few studies on software–hardware coordinated control and the hardware control strategy. After the control command is issued, the switch will not act immediately due to the influence of the working characteristics, and there is a delay effect on the time scale, which causes the quality of the power supply to decline and even threatens the safety of the local load.

### 1.4. Contribution and Scope

To solve the existing problems in current research, this paper proposes a new type of SSCT to solve the problem of sudden voltage changes in the process of GCM and OGM. The main contributions of this paper can be summarized as follows:

(1) Aiming at the zero-crossing shutdown characteristics of SCR, the SCR-FSCS is proposed, which significantly compresses SCR turn-off time. This strategy avoids sudden changes in the load voltage.

(2) This paper proposes to adopt a TLCS to realize the rapid conversion between GCM and OGM. This strategy can carry out real-time control of the current, has a fast response speed, and provides software guarantee for SSCT.

(3) This paper uses the GCC platform to carry out related experiments, such as SCR switching characteristics, the influence of SCR switching characteristics on the load voltage, and the SSCT of GCC different load types.

### 1.5. Organization of the Document

The other parts of the paper are arranged as follows. In Section 2, the typical results of MG are introduced. In Section 3, the preconditions for SSCT of GCC are introduced. In Section 4, the influence of the switching characteristics of SCR on the local load and the specific realization method of the SCR-FSCS are analyzed. In Section 5, the design method of the TLCS is introduced and the stability of TLCS is analyzed. In Section 6, combining the SCR-FSCS and the TLCS, the specific implementation method of the proposed SSCT is introduced. In Section 7, the effect of SSCT is verified through experiments. In Section 8, the main conclusions of this paper are given.

## 2. Topology of the MG and GCC

The typical MG topology is shown in Figure 1. Photovoltaic cells are connected to DC bus through a DC converter. The DC bus provides power to the local sensitive load through a GCC. The GCC is connected to the power grid through a GCC, circuit breaker, and transformer to realize energy exchange with the power grid. The energy storage system is connected to the DC bus through a bidirectional DC/DC converter.

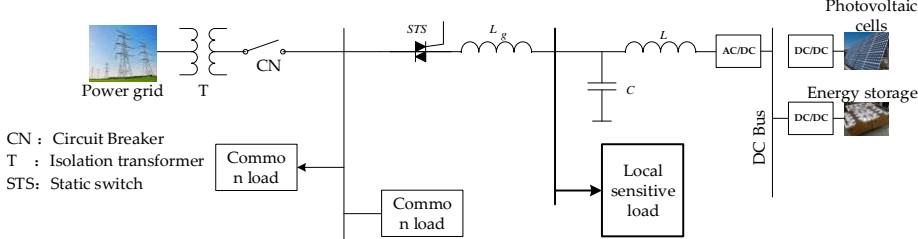

**Figure 1.** Topological structure diagram of the micro-grid (MG).

Considering the need for unbalanced loads, a three-phase four-wire topology based on a flying capacitor is adopted for the GCC, and an LCL filter is also applied, as shown in Figure 2. Two filter capacitors are connected in a series on the DC side to obtain a stable DC bus voltage. The midpoint of the two capacitors is taken as the midpoint of the system output voltage. The topology includes three single-phase half bridges, so the circuit can work in a three-phase state and single-phase state. Because the three-phase control of the topology is realized independently, it has a strong ability, with an unbalanced load.

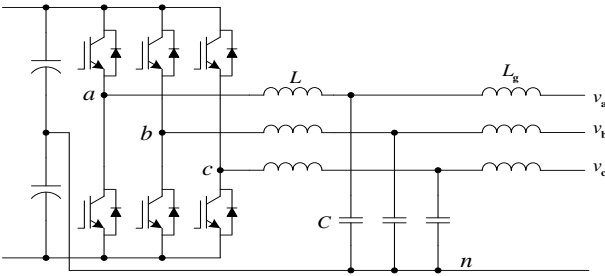

**Figure 2.** Topological structure diagram of a three-phase four-wire converter based on a flying capacitor.

## 3. Preconditions for the Seamless Switching of a GCC

In the ideal state, the power grid supplies a constant frequency, sinusoidal waveform, and standard voltage to the load. However, due to the influence of various power quality problems, so the ideal waveform does not exist. In the MG, a large number of distributed renewable energy power generation units are connected to the grid, making the power quality problem of the grid more serious. To ensure the power safety of the load, the switching of the GCC from GCM to OGM needs to meet the following conditions:

(1) The fluctuation range $\Delta U$ of the grid voltage amplitude of any one phase is not more than $\pm$ x%.
(2) The time $\Delta t$ taken to switch from the GCM to the OGM should be short enough to ensure uninterrupted power supply to the load, which can be set as $\Delta t < z$.

When the grid voltage recovers from the fault, the GCC should switch from the OGM to the GCM. During the grid-connection process, it is necessary to ensure that no impulse current is generated, and a smooth transition of the load voltage is realized. Therefore, the following conditions should be met for the switching of the GCC from the OGM to the GCM:

(1) The grid voltage is normal.
(2) The Voltage phase difference $\Delta\theta$ between the grid voltage and the load voltage is small enough, not exceeding $\pm\varepsilon$.
(3) The Voltage amplitude difference $\Delta U_1$ between grid voltage and load voltage is small enough, not exceeding $\pm\sigma$.

A logic diagram for the seamless switching of the GCC is shown in Figure 3, which includes the control command, operation mode switching, and control effect. The control commands can be sent by

the upper management system or the local controller. Both of them manage the operation mode of the GCC through the logic, "or". The operation mode includes two steady-state operation modes of OGM and GCM, as well as their switching process and switching conditions. The control effect is the control goal of the whole switching process, that is, in the switching process, the continuity of load voltage-time should be ensured, and instantaneous power interruption should not occur. When this occurs, the switching process is called seamless. At the same time, the waveform quality of the local load voltage should be ensured, that is, the amplitude and phase should not have large fluctuations.

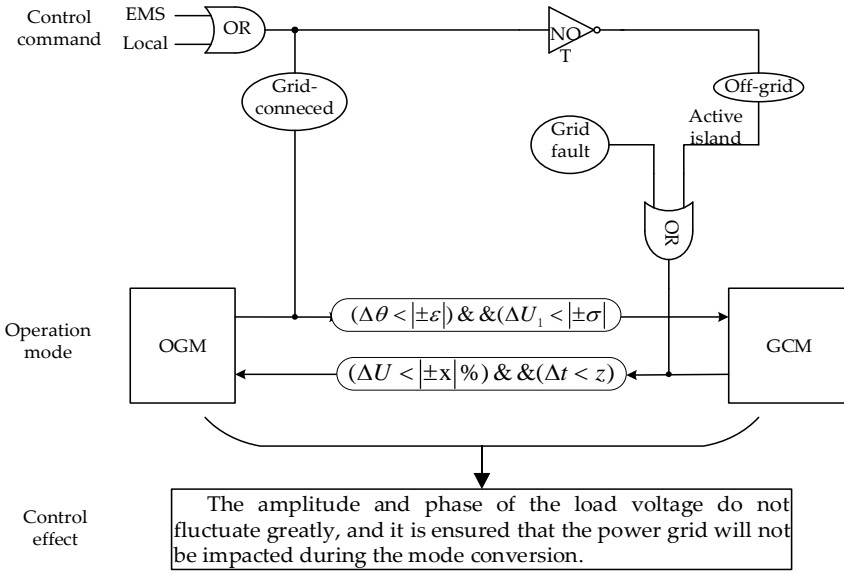

**Figure 3.** Logic diagram for the seamless switching of the grid-connected converter (GCC).

## 4. Control Strategy of the GCCS

### 4.1. Operation Characteristic Analysis of the GCCS

The GCC needs to be connected to the grid through switch equipment. The switch equipment must have the ability to allow large currents to pass and turn off reliably. Therefore, in the selection of switch equipment, the action time and current carrying capacity need to be considered. As shown in Figure 4, SCR is the ideal choice for a grid-connected control switch.

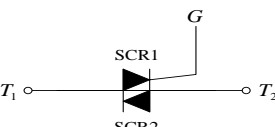

**Figure 4.** The silicon-controlled rectifiers (SCR) switch.

It can be seen, from Figure 4, that since two cells in reverse parallel are integrated into the die, the SCR switch can be controlled by one gate, G, in turn. When the GCC is connected to the grid, a high-level signal is applied to the gate. Within the half cycle of the grid voltage, there is one SCR inside each switch that bears the forward bias voltage, that is, the two SCRs turn on alternately to realize the connection control of the MG and the grid. However, SCR can only be controlled by a gate signal, but it cannot be turned off. When the gate trigger signal is removed, the SCR will not be turned off immediately. The off time depends on the time when the gate trigger signal is removed. As shown in Figure 5, after the trigger signal is removed, the three-phase current through SCR is turned off in turn.

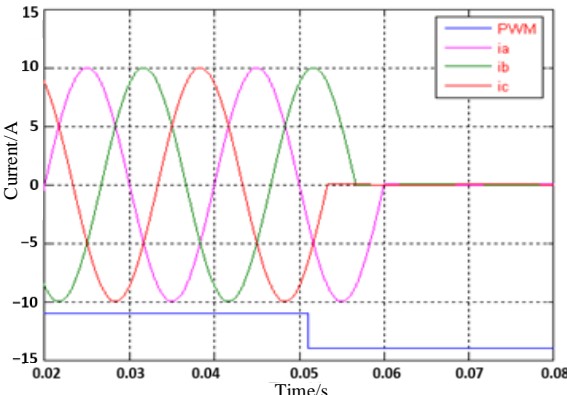

**Figure 5.** Current waveform of the SCR, when it is free to turn-off.

The essence of the switching control that allows the GCC to switch from the GCM to the OGM is the ability to switch from the current source to the voltage source. Only when the three-phase SCR is completely turned off can the GCC switch from the GCM to the OGM. Therefore, when the three-phase SCR starts to turn completely turn off, the GCC is still in the current source mode. As shown in Figure 6a, taking a single phase as an example, when the energy emitted by the MG exceeds the load demand, the excess energy is injected into the grid. Before SCR disconnection, the following equation holds:

$$i_1 = i_{l,o,a,d} + i_g \tag{1}$$

where $i_1$ is the output current of the GCC; $i_{l,o,a,d}$ is the load current, and $i_g$ is the grid current, because

$$i_1 \geq i_{l,o,a,d} \tag{2}$$

$$v \approx e \tag{3}$$

where $v$ is the voltage of load, and $e$ is the voltage of grid.

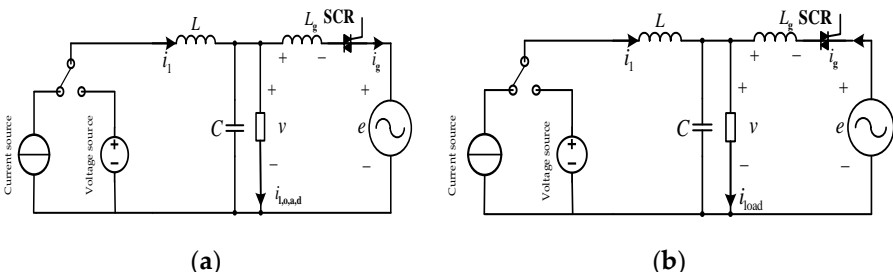

**(a)**             **(b)**

**Figure 6.** Schematic diagram of the mode switching of the GCC. (**a**) Working state 1. (**b**) Working state 2.

Equation (4) can be obtained from Equations (1) and (3):

$$v = i_{l,o,a,d} * R = (i_1 - i_g) * R \tag{4}$$

where $R$ is the local load.

When the SCR of this phase is disconnected, the GCC still adopts the current source control mode because the SCRs of other two-phase has not been completely disconnected. At this time, the following equations are valid:

$$i_{l,o,a,d} = i_1 \tag{5}$$

$$v = i_{l,o,a,d} * R = i_1 * R \tag{6}$$

It can be seen from Equations (1), (3), and (6) that

$$v > e \tag{7}$$

$$\Delta v = i_{\mathrm{g}} * R \tag{8}$$

where $\Delta v$ is voltage fluctuation amplitude.

It can be seen, from Equations (7) and (8), that the local voltage of this phase will rise abruptly after the SCR of a certain phase is first turned off, and the amplitude of the sudden rise in voltage is linearly related to the grid-connected current. When the voltage amplitude rises to a certain extent and exceeds the regulation range of the converter, the converter will be oversaturated, and the load voltage amplitude will become half of the DC bus voltage. Such a high voltage will threaten the power supply safety of the load.

As shown in Figure 6b, when the energy emitted by the MG is less than the load demand, insufficient energy is provided by the grid. Under this condition, when the SCR is not disconnected, the following equations are valid:

$$i_{\mathrm{l,o,a,d}} = i_1 + i_{\mathrm{g}} \tag{9}$$

$$i_{\mathrm{l,o,a,d}} \geq i_1 \tag{10}$$

$$v = i_{\mathrm{l,o,a,d}} * R = (i_1 + i_{\mathrm{g}}) * R \tag{11}$$

When this phase SCR is disconnected, the other two-phase SCR is not completely disconnected, and the GCC still adopts the current source control mode. At this time, the following equations are established:

$$i_{\mathrm{l,o,a,d}} = i_1 \tag{12}$$

$$v = i_{\mathrm{l,o,a,d}} * R = i_1 * R \tag{13}$$

It can be seen from Equations (9), (10), and (13) that

$$v < e \tag{14}$$

$$\Delta v = -i_{\mathrm{g}} * R \tag{15}$$

It can be seen, from Equations (14) and (15), that in the process of the GCC switching from the GCM to the OGM, when the SCR is not completely turned off, the phase voltage will sag after the SCR of a certain phase is first turned off. The amplitude of the voltage sag is linearly related to the grid current, and the voltage sag will affect the power supply stability of the local load.

*4.2. The Control Strategy of Forcing the SCR to Shutdown*

During the time when the three-phase SCR is turned off sequentially, the problem of a sudden rise or sag of the load voltage will occur. Therefore, the turn-off time of SCR must be shortened as much as possible to prevent sudden rise or sag in load voltage. A converter circuit is usually added to apply reverse voltage, which allows the current flowing through SCR to be adjusted, and the conductive SCR is forced to turn off quickly. However, adding a converter circuit will increase the volume and cost of the GCC and the coordination control difficulty. Therefore, this paper proposes a novel SCR-FSCS, without adding any converter circuit. The SCR-FSCS is completely implemented by digital programs, forcing the three-phase SCR to be turned off before the load has over-voltage or under-voltage.

As shown in Figure 7, the grid-connected current $i_{\mathrm{g}}$ is injected into the power grid through the $L_{\mathrm{g}}$ and SCR. When the SCR trigger signal changes from high to low, as long as the counter voltage is formed in the inductance $L_{\mathrm{g}}$, the SCR can be turned off quickly. Since the grid voltage is relatively fixed, the core idea of the SCR-FSCS is to adjust the load side voltage to form a back voltage on $L_{\mathrm{g}}$. The back voltage forces $i_{\mathrm{g}}$ to decrease rapidly, which makes the SCR quickly change from the on

state to the blocking state, thus breaking the connection with the grid. Because the grid voltage is an alternating current, the amplitude and phase angle change in real time. To force the SCR to shut down, it is necessary to study how to change the load voltage to form back voltage at both ends of $L_g$, as well as the factors affecting the off time.

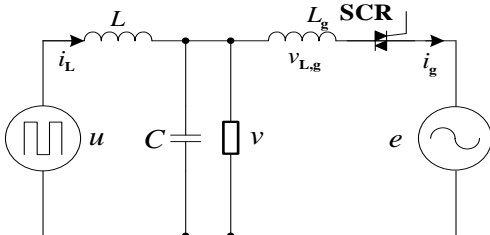

**Figure 7.** The single-phase grid connection diagram of GCC.

As shown in Figure 7, according to the Kirchhoff voltage law, the following equation holds:

$$v_{L,g} = v - e \tag{16}$$

where $e$ is the grid voltage; $v_{L,g}$ is the voltage of $L_g$, and $v$ is the load voltage.

The phase $e$ is set to 0 and the amplitude to $E$, and the following equation holds:

$$e = E \sin \omega t \tag{17}$$

The phase of $i_g$ is set as $\alpha$, and the amplitude is $I_g$, the following equation is valid:

$$i_g = I_g \sin(\omega t + \alpha) \tag{18}$$

The phase of $v$ is set as $\theta$, and the amplitude is $V$, and the following equation holds:

$$v = V \sin(\omega t + \theta) \tag{19}$$

From Equations (16)–(19), it can be seen that:

$$\begin{cases} V = \sqrt{E^2 - 2wL_g I_g E \sin \alpha + \omega^2 L_g^2 I_g^2} \\ \theta = \arctan \frac{\omega L_g I_g \cos \alpha}{E - w L_g I_g \sin \alpha} \end{cases} \tag{20}$$

The voltage $v_{L,g}$ at both ends of $L_g$ is

$$v_{L,g} = V \sin(\omega t + \theta) - E \sin \omega t \tag{21}$$

After the change of the load voltage, the amplitude is set as $V'$, and let $V' = xV$. To facilitate the theoretical analysis, $x \in (0, 1) \cup (1, 2)$, and the voltage $V_{L,g}'$ at both ends of $L_g$ is

$$v_{L,g}' = xV \sin(\omega t + \theta) - E \sin \omega t \tag{22}$$

A large actual load voltage fluctuation should not occur, and the value range of $x$ should be determined according to the tolerance of the local load to voltage fluctuation. In this paper, the voltage fluctuation range is not more than 20%. That is, $x \in (0.8, 1) \cup (1, 1.2)$.

When the grid-connected current drops to 0, and the time $\Delta t$ is short enough, the following formula holds:

$$\Delta t = L_g \frac{\Delta i_g}{\Delta v_{L,g}} = L_g \frac{0 - i_g}{v_{L,g}' - v_{L,g}} \tag{23}$$

where $\Delta i_g$ is the fluctuation amplitude of $i_g$, and $\Delta v_{L,g}$ is the fluctuation amplitude of $v_{L,g}$.

By introducing Equations (21) and (22) into Equation (23), we can obtain the following results:

$$\Delta t = L_g \frac{I_g \sin(\omega t + \alpha)}{(1-x)V \sin(\omega t + \theta)} \tag{24}$$

According to Equation (24), there are two cases, which are as follows:

(1) When $i_g \cdot v > 0$, i.e., $i_g$ and $v$ are in phase, the amplitude of the load voltage was reduced to make it lower than $e$.

(2) When $i_g \cdot v < 0$, i.e., $i_g$ and $v$ are in the inverse phase, and the amplitude of the load voltage is increased to make it higher than $e$.

As shown in Figure 8, in regions $t_1$ and $t_3$, the two are in an the inverse phase, and the amplitude of $v$ needs to be increased; in areas $t_2$ and $t_4$, both are in phase, and the amplitude of $v$ needs to be decreased.

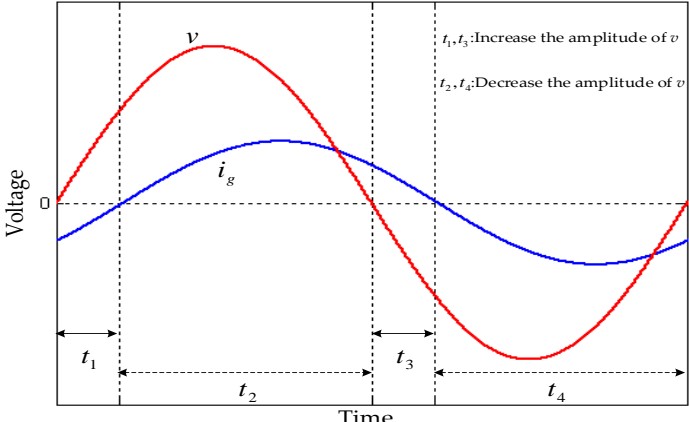

**Figure 8.** Schematic diagram of capacitor voltage amplitude adjustment area.

The sine component in Equation (24) is ignored, because the time is short enough:

$$\Delta t = \begin{cases} L_g \frac{I_g}{V(1-x)} x \in (0,1) \\ L_g \frac{I_g}{V(x-1)} x \in (1,2) \end{cases} \tag{25}$$

The maximum $I_{g,m,a,x}$ is determined by ground short-circuit current of the load voltage as shown in Equation (26).

$$I_{g,m,a,x} = \frac{V}{\omega L_g} \tag{26}$$

It is assumed that $I_g$ is $y$ times the maximum value $I_{g,m,a,x}$, i.e., $I_g = y I_{g,m,a,x}$, where $y < 1$. From Equations (25) and (26), we can get

$$\Delta t = \begin{cases} \frac{y}{\omega(1-x)} x \in (0,1) \\ \frac{y}{\omega(x-1)} x \in (1,2) \end{cases} \tag{27}$$

Figure 9 can be obtained from Equation (27); according to the size of $I_g$, the time that it takes for the grid-connected current to drop to 0 in a different range of load voltage amplitude can be obtained. When the amplitude of the load voltage changes from 10% to 20%, the time that it takes for the grid-connected current to drop to 0 is within 3 ms, which is significantly shorter than when the SCR is free to turn off. When the amplitude of the load voltage changes in a certain range, the turn-off time

of the SCR depends on $I_g$. The larger the grid current is, the longer the turn-off time is. It should be emphasized that, in Figure 9, when $x = 1$, the turn off time $\Delta t$ is infinite. In fact, in Equation (27), $x = 1$ is not in the defined region. When $x = 1$, the scope of the strategy of forcing the SCR to shutdown is beyond the defined region, and the SCR is free to turn off.

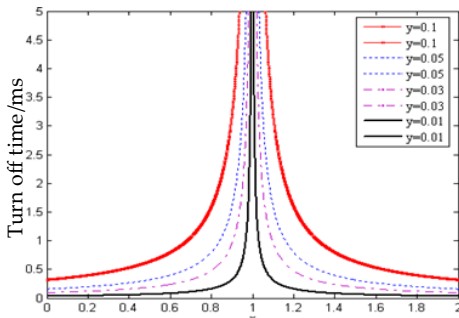

**Figure 9.** The change curve of the off time corresponding to the capacitor voltage and grid-connected current.

## 5. The Control Strategy of GCC

### 5.1. The Novel Three-Loop Control Strategy for GCC

As shown in Figure 2, Equation (28) can be obtained from the Kirchhoff voltage law for the grid-connected inductor $L_g$.

$$\begin{cases} L_g \frac{di_{g,a}}{dt} = v_a - e_a \\ L_g \frac{di_{g,b}}{dt} = v_b - e_b \\ L_g \frac{di_{g,c}}{dt} = v_c - e_c \end{cases} \tag{28}$$

where $i_{g,a}$, $i_{g,b}$, and $i_{g,c}$ are the three phases current of $L_g$; $v_a$, $v_b$, and $v_c$ are the three phases load voltage; and $e_a$, $e_b$, and $e_c$ are the power grid voltage. Park transform was used to obtain the following results.

$$\begin{cases} v_d = L_g \frac{di_{g,d}}{dt} + \omega L_g i_{g,q} + e_d \\ v_q = L_g \frac{di_{g,q}}{dt} - \omega L_g i_{g,d} + e_q \end{cases} \tag{29}$$

In the dq coordinate system, $v_d$ and $v_q$ are the components of $v$ in the d and q axes; $i_{g,d}$ and $i_{g,q}$ are the components of the $i_g$ in the d and q axes; $e_d$ and $e_q$ are the components of $e$ in the d and q axes.

The voltage of the grid-connected inductor is defined as Equation (30):

$$\begin{cases} v_{L,g,d} = L_g \frac{di_{g,d}}{dt} + \omega L_g i_{g,q} \\ v_{L,g,q} = L_g \frac{di_{g,q}}{dt} - \omega L_g i_{g,d} \end{cases} \tag{30}$$

As shown in Figure 10, It can be seen from Equation (30) that the real-time control of the $v_{L,g}$ can be realized by the direct control of $i_g$, which is completed by the current controller. Then according to Equation (30), the feedforward of the grid voltage $e_{d,q}$ is introduced, and their sum is set as the voltage of the voltage controller $e^*_{d,q}$. Then, the inductor current control at the converter side is introduced to form a TLCS for the GCC, as shown in Figure 10. The control strategy maintains a good continuity with the dual-loop control system in independent operation, which is useful for realizing the seamless switching control of the GCC.

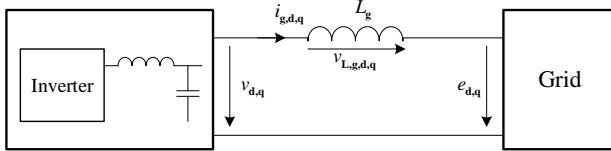

**Figure 10.** Schematic diagram of the connection between the GCC and power grid.

The TLCS is shown in Figure 11. It is an optimized combination of voltage source control and current source control, which has the advantages of both. It can directly control the grid-connected current in real-time, with a fast response speed, and it is not affected by system parameter changes and grid voltage fluctuations. When the GCC changes from GCM to OGM, the back voltage can be formed at both ends of $L_g$ by changing the given value of the voltage loop. The SCR-FSCS proposed in the previous section can be realized to make SCR turn off quickly and complete the mode switching of the GCC. When the GCC changes from the OGM to the GCM, the given voltage loop value can be adjusted in real-time to keep it consistent with the grid voltage all the time and avoid the impact of the grid-connected current during mode switching.

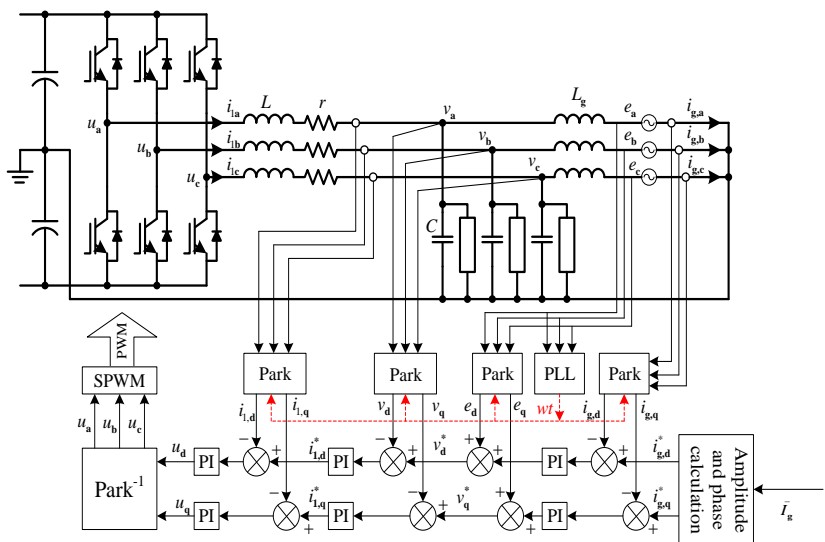

**Figure 11.** Control block diagram of three-loop control strategy (TLCS).

### 5.2. Design and Stability Analysis of the New Three-Loop Control Strategy

After the GCC adopts the TLCS, it is a high-order system. In order to reduce the difficulty of control parameter design, the strategy is designed by using the equation of state in the synchronous rotating coordinate system. The specific flow is shown in Figure 12. Firstly, the state space equations of the open-loop and three-loop control of the GCC are established and linearized. Then, the closed-loop control state equation of the GCC is integrated. According to experience, the three-loop control parameters are selected, and stability analysis is carried out. If the system stability meets the requirements, this group of parameters will be selected; otherwise, the next group of control parameters will be selected, and stability analysis will be carried out before the selected parameters will be considered to have met the stability requirements.

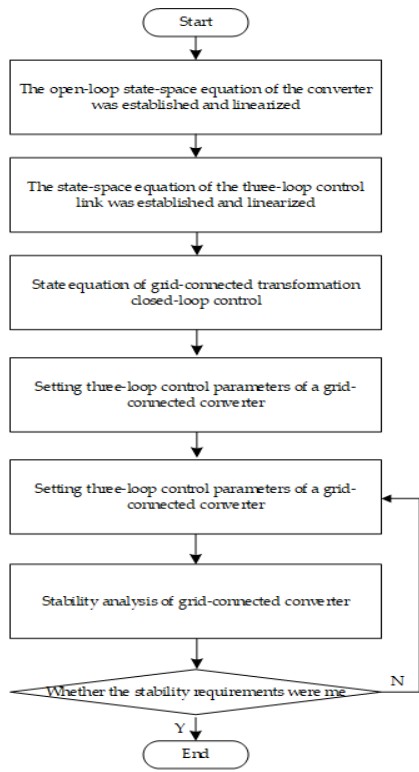

**Figure 12.** Flow chart of control parameter design for GCC.

The stability of the GCC is analyzed, and the open-loop state-space equation and control link state-space equation of the GCC are established, as shown in in Appendix A. Moreover, the meaning of each variable is shown in Table A1 in Appendix A. The state matrix of the GCC can be obtained by linearization and integration, as shown in Equation (A3) in Appendix A. The eigenvalues of the GCC can be obtained using Equation (A3). As shown in Table A2, the real part of each eigenvalue is negative. It is far away from the imaginary axis, which had a sufficient stability margin. This indicated that the selected control parameters can meet the control and stability requirements of the GCC.

## 6. Seamless Switching Control of GCC

### 6.1. Control Technology for Switching from the GCM to the OGM of GCC

Combined with the SCR-FSCS and TLCS, the control technology for switching the GCC from the GCM to the OGM is proposed. As shown in Figure 13, 0 represents the OGM, and 1 represents the GCM. The states of the grid voltage are represented by 0 and 1, 0 representing fault and 1 representing a normal state. When the GCC works in the GCM, the SCR switch trigger signal is high. The $S_2$ selects GCM 1. When the grid fails or the operation command is changed to the OGM, the trigger signal of the SCR switch changes to a low level, and the mode selection switch $S_2$ selects the OGM 0. The off-grid control strategy of the GCC is the load voltage outer loop and inductor current inner loop. According to SCR-FSCS, when $S_2$ selects mode 0, the given load voltage $v^*_{d,q}$ changes from $(k_p + k_i/s)(i^*_{g,d,q} - i_{g,d,q}) + e_{d,q}$ to $x \cdot e_{d,q}$. A back voltage is formed at $L_g$. The GCC completes the conversion from the GCM to OGM. At this time, the load voltage setting selection switch $S_1$ is changed from $x \cdot e_{d,q}$ to $v_{d,q,r,e,f}$, and the load voltage quickly returns to a normal level. The phase of $e$ when in the OGM must be considered in connection with the given value of the load voltage $v_{d,q,r,e,f}$. If the phase of $e$ when in the of OGM is set as $\theta$, then

$$\begin{bmatrix} v_{d,r,e,f} \\ v_{q,r,e,f} \end{bmatrix} = \begin{bmatrix} V\cos\theta \\ -V\sin\theta \end{bmatrix} \tag{31}$$

where *V* is the rated amplitude of load voltage, and *θ* is the initial phase.

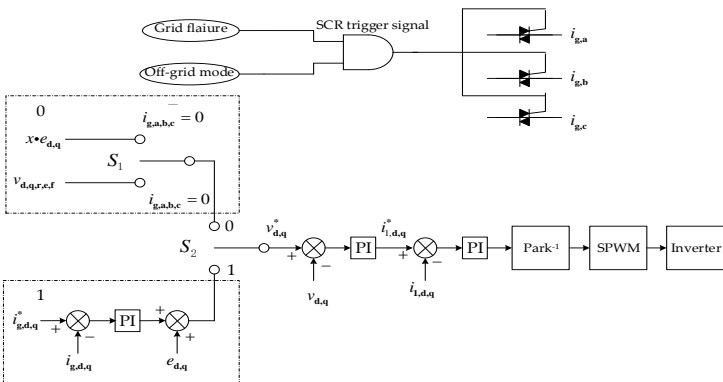

**Figure 13.** Logic diagram: from the grid-connected mode (GCM) to the off-grid mode (OGM).

*6.2. Control Technology for Switching from OGM to the GCM*

When the grid voltage returns to normal, GCC needs to be re-connected to the grid. The control strategy conversion and SCR connection should be realized at the same time to ensure that the SCR will not produce a grid-connected current impact at the moment of connection and realize a smooth transition of the load voltage.

$$i_\mathrm{g} = \frac{v - e}{j\omega L_\mathrm{g}} \tag{32}$$

The grid-connected current can be expressed by Equation (32). The grid-connected current value at the moment of grid-connection depends on the difference between *v* and *e*. The equivalent impedance of the $L_\mathrm{g}$ is very small, and the slight will cause a great impact on $i_\mathrm{g}$. Therefore, when the GCC is switched from the OGM to the GCM, *v* should first be adjusted to allow it to gradually synchronize with *e*. Unlike the strict requirements for the conversion time, when switching from GCM to OGM is carried out, the conversion time is not strictly limited to switching from OGM to GCM. This is mainly due to the need for maintaining a smooth transition of the load voltage waveform and phase.

The control strategy of the GCC from OGM to GCM is shown in Figure 14. Mode 0 represents the OGM, and mode 1 represents GCM. A 0 command represents a fault, and a 1 command represents a normal power grid. When the GCC receives the grid-connection command, it will gradually adjust the given load voltage $v_\mathrm{d,q,r,e,f}$ to track the grid voltage step-by-step. When the error is small enough, a high-level trigger signal will be send to the SCR, and $S_2$ changes to GCM 1. The GCC control strategy is changed from the double-loop voltage source control to three-loop current source control strategy. The GCC turns into a grid-connected operation mode. It should be noted that we would rather make the tracking time of t*v* to *e* longer, rather than shorter, because the phase should be adjusted slightly to prevent the phase-sensitive load from being affected. The given value $v_\mathrm{d,q,r,e,f}$ of the load voltage can be expressed by Equation (31).

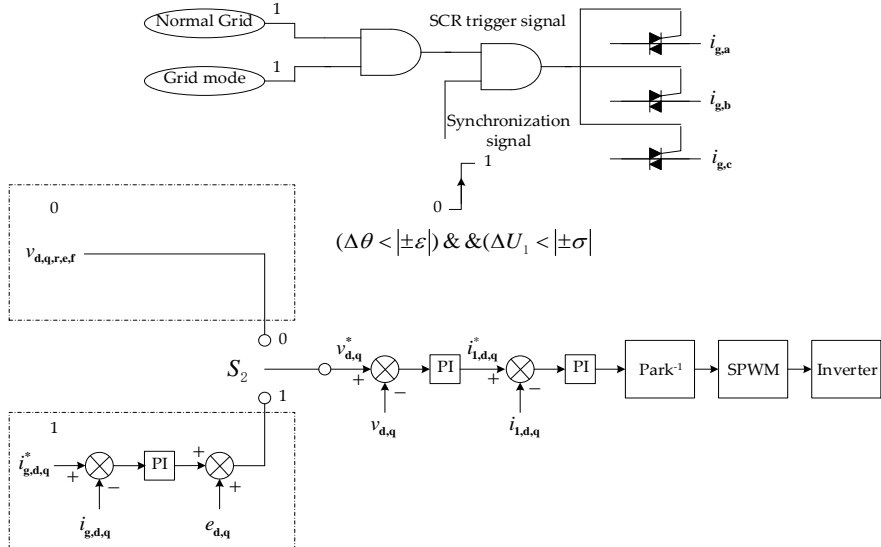

**Figure 14.** Logic diagram: from the OGM to the GCM.

## 7. Experimental Research of GCC

As shown in Figure 15, a 100 kW GCC is built for the relevant experimental verification. The main control chip model of the GCC is TMS320F28335 (Texas Instruments, Dallas, TX, USA). The SCR model is MCC162-16io1 (IXYS Corporation, Milpitas, CA, USA), and Table 1 shows the specific parameters. Table 2 shows the main parameters of the GCC.

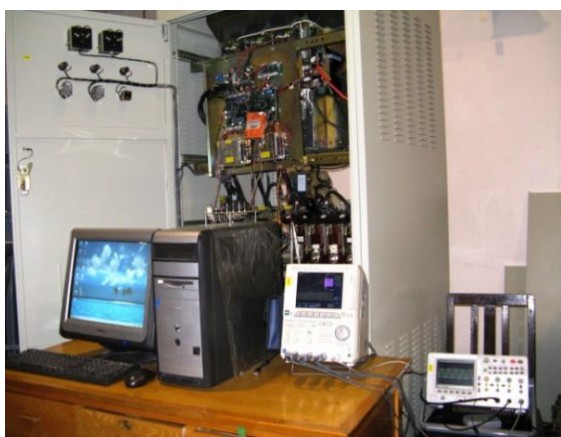

**Figure 15. The** GCC experimental platform.

**Table 1. The** parameters of SCR switch.

| Parameters | Numerical Value |
| --- | --- |
| Forward blocking voltage/V | 1600 |
| Forward rated current/A | 180 |
| Conduction voltage drop/V | 1.03 |

**Table 2.** The parameters of GCC.

| Parameters | Numerical Value |
| --- | --- |
| Rated phase voltage at AC side/V | 220 |
| The rated voltage at DC side/V | 750 |
| Rated output power/kW | 100 |
| Rated frequency/Hz | 50 |
| Switching frequency/kHz | 6 |
| LCL wave filter inductance at machine side/mH | 0.6 |
| LCL wave filter inductance at grid side/mH | 0.5 |
| LCL filter capacitor/$\mu$F | 60 |
| Outer loop parameters of grid-connected current | Kp = 0.3; Ki = 0.313 |
| Middle loop parameters of local load voltage | Kp = 0.125; Ki = 0.0078 |
| Current inner loop parameters of filter inductor | Kp = 0.25; Ki = 0.313 |

*7.1. Experimental Analysis of Operating Characteristics of the SCR*

The parameters of SCR are shown in Table 1. It is connected to the three-phase circuit for the on-off test, as shown in Figure 16. When the SCR trigger pulse achieves a low level, the three-phase current will not be turned off immediately, rather, it will turn off naturally with the zero-crossing point of each phase current. The time taken to turn off the three-phase current depends on the time it takes for the driving pulse to achieve a low level.

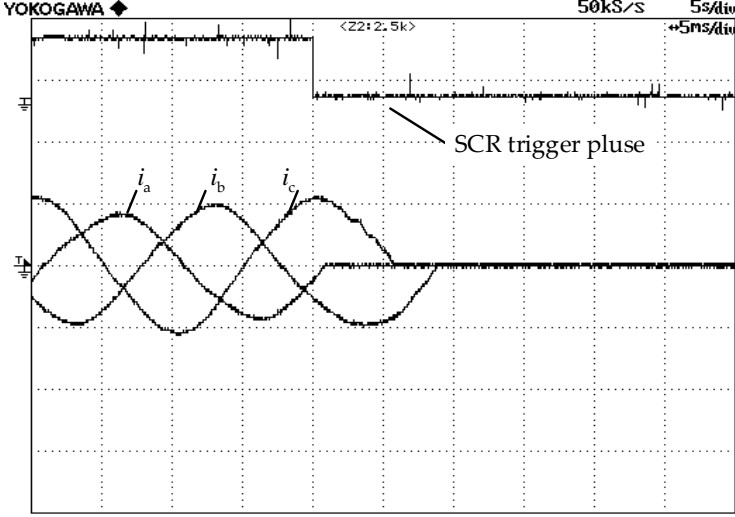

**Figure 16.** SCR, when it is free to turn off the waveform.

When the GCC adopts the current source grid-connected control strategy, the SCR trigger pulse achieves a low level, and the voltage of load experimental waveform is shown in Figure 17. If the GCC injects energy into the grid at this time, when SCR trigger signal achieves a low level, the load voltage will jump. If the GCC absorbs energy from the grid and supplies power to the load together, when the SCR trigger signal achieves a low level, the load voltage will sag. A voltage surge or sag will affect the power quality of the load voltage, which proves the rationality and correctness of the theoretical analysis in this paper.

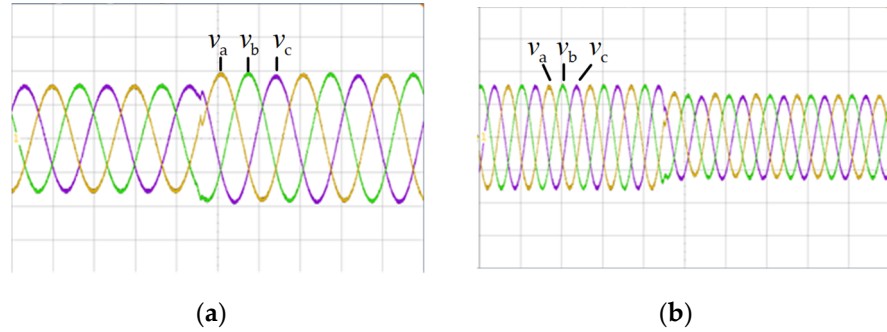

**Figure 17.** Load voltage waveform when the SCR is turned off. (**a**) Working state 1 (energy is injected into the power grid). (**b**) Working state 2 (energy from the power grid).

### 7.2. Experimental Study of Switching the GCC from the GCM to the OGM

The experimental waveforms of the load voltage and grid current are shown in Figure 18. When the SCR trigger signal changes from high to low, load voltage still exists. Its phase is also consistent with that during grid connection, thus ensuring the continuity and reliability of the load power supply. Under the action of SCR-FSCS, all three phases of the grid-connected currents are rapidly reduced to zero, and the GCC then converts to the OGM. The grid-connected current, total harmonic current distortion (THDi), is shown in Figure 19. It is 3.0%, thus, meeting the grid-connection standard.

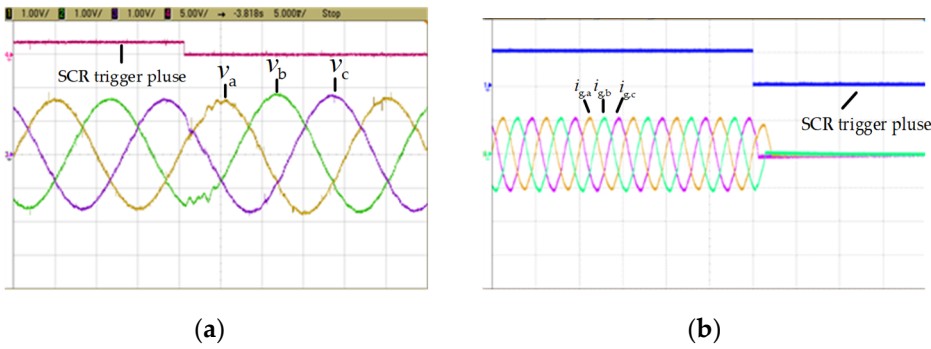

**Figure 18.** The waveform of the GCC changes from the GCM to the OGM. (**a**) Experimental waveform of the load voltage. (**b**) Experimental waveform of the grid current.

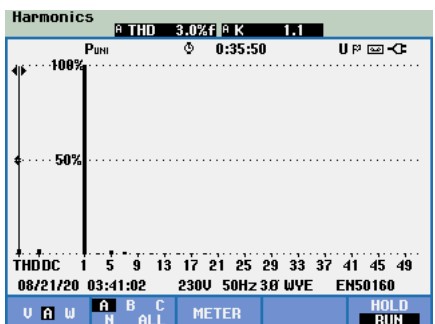

**Figure 19.** The THDi of the grid current.

### 7.3. Experimental Study on the GCC Switching from the OGM to the GCM

The experimental waveform of the GCC switching from the OGM to the GCM is shown in Figure 20. The SCR trigger signal changes from low to high. At the moment of grid connection, the given value of the local load voltage changes from $v_{d,q,r,e,f}$ to $(k_p + k_i/s)(i^*_{g,d,q} - i_{g,d,q}) + e_{d,q}$. According to the experimental waveform, the voltage of load does not change significantly during the switching process,

and the grid-connected current increases gradually, which indicates that the proposed control of the switching from the OGM to GCM achieved a good control effect. The grid-connected current, THDi, is shown in Figure 21. It is 3.2%, thus meeting the grid-connection standard. The effectiveness and rationality of the strategy are therefore proved.

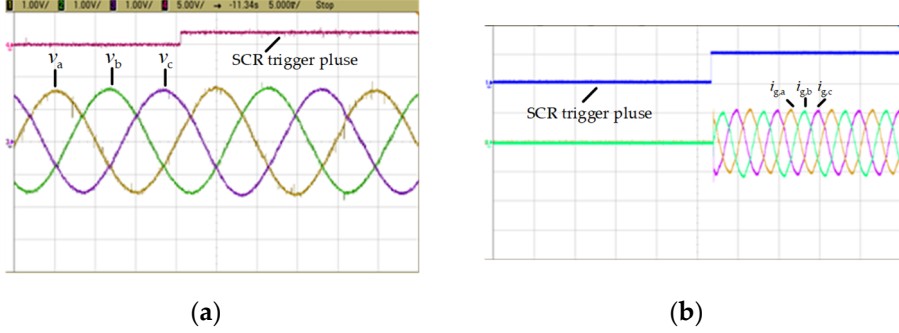

(**a**)　　　　　　　　　　　　　　　　　　　　　(**b**)

**Figure 20.** The waveform of GCC changes from the OGM to the GCM. (**a**) Load voltage waveform. (**b**) Grid-connected current waveform.

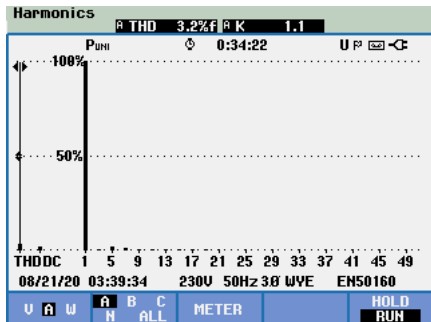

**Figure 21.** The THDi of the grid-connected current.

### 7.4. Experimental Study on the GCC with a Nonlinear Load

When the GCC switches seamlessly, the experimental waveform of the nonlinear load is shown in Figure 22. When the GCC changes from the OGM to the GCM, the SCR trigger signal changes from low to high, and the nonlinear load current remains continuous, which indicates that the load power supply remained continuous during the switching. The THDi of grid-connected current is shown in Figure 23. It is 3.9%, thus meeting the grid-connection standard.

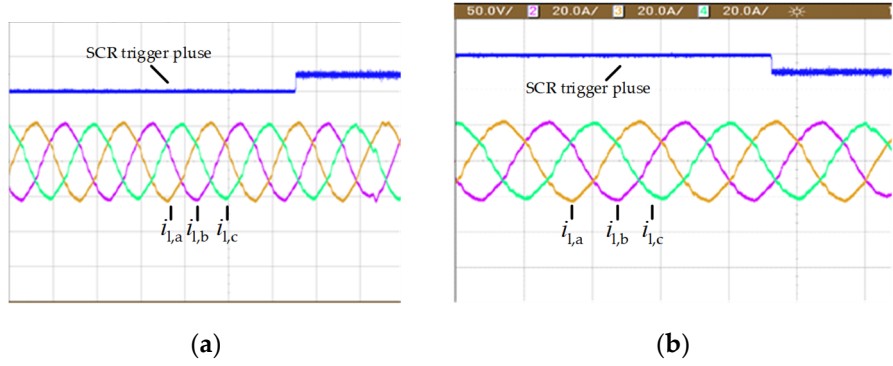

(**a**)　　　　　　　　　　　　　　　　　　　　　(**b**)

**Figure 22.** Nonlinear load current waveform during the seamless switching of the GCC. (**a**) Nonlinear load current waveform (from the OGM to the GCM). (**b**) Nonlinear load current waveform (from the GCM to the OGM).

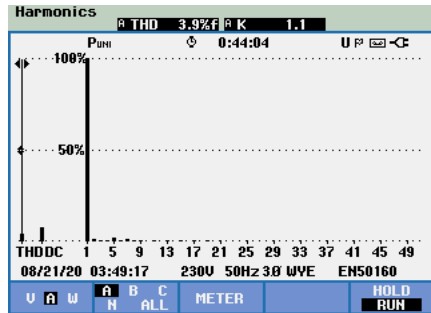

**Figure 23.** The THDi of the nonlinear load current.

## 8. Conclusions

In this paper, on the basis of considering the characteristics of the SCR, the SSCT for GCC is proposed. Through the research work carried out in this paper, the following conclusions can be drawn:

(1) Affected by the zero-crossing shutdown characteristics of the SCR, when the GCC in the MG switches between the GCM and the OGM, the load voltage will suddenly drop or rise, which will affect the reliability of the local load.

(2) The turn-off time of SCR is closely related to the local load voltage and grid-connected inductor current. The SCR-FSCS greatly reduces the turn-off time of the SCR by forming a back voltage at both ends of the grid-connected inductor to ensure the stability of the load power supply.

(3) The TLCS can easily switch between the current source three-loop control mode and the voltage source double-loop control mode. At the same time, the stability of the TLCS is analyzed, which provides a reference for the selection of key parameters of the control strategy.

Through the research of this paper, it can be known that after the GCC adopts the proposed SSCT, the reliability of the local load power supply is guaranteed. However, when the load is a nonlinear load, the THDi of the current is still large. Therefore, the focus of future work is to improve the adaptability of the strategy and reduce the THDi of different types of loads.

**Author Contributions:** Conceptualization, C.S. and T.W.; methodology, C.S.; software, C.S.; validation, C.S. and Y.S.; formal analysis, C.S.; investigation, D.J.; data curation, T.L. and Y.S.; writing—original draft preparation, C.S.; writing—review and editing, T.W.; supervision, D.J.; project administration, T.L. All authors have read and agreed to the published version of the manuscript.

**Funding:** This research was funded by the National Key Research and Development Program of China (2016YFB0900400), the Strategic Priority Research Program of the Chinese Academy of Sciences (XDA01020304) and Key Research and Development Program of Hebei Province (19214504D).

**Conflicts of Interest:** The authors declare no conflict of interest.

## Appendix A

$$\begin{cases} \frac{di_{1,d}}{dt} = -\frac{R}{L}i_{1,d} - \omega i_{1,q} + \frac{1}{L}(v_d - v_{c,d}) \\ \frac{di_{1,q}}{dt} = -\omega i_{1,d} - \frac{R}{L}i_{1,q} + \frac{1}{L_c}(v_q - v_{c,q}) \\ \frac{dv_d}{dt} = \frac{1}{C}(i_{g,d} - i_{1,d}) + \omega v_q \\ \frac{dv_q}{dt} = \frac{1}{C}(i_{g,q} - i_{1,q}) - \omega v_d \\ \frac{di_{g,d}}{dt} = -\frac{R_g}{L_g}i_{g,d} - \omega i_{g,q} + \frac{1}{L_g}(v_{s,d} - v_d) \\ \frac{di_{g,q}}{dt} = -\frac{R_g}{L_g}i_{g,q} + \omega i_{g,d} + \frac{1}{L_g}(v_{s,q} - v_q) \end{cases} \tag{A1}$$

$$
\begin{cases}
d_{\mathrm{N,d}} = d_{\mathrm{N,d,p}} + d_{\mathrm{N,d,i}} \\
d_{\mathrm{N,d,p}} = k_{\mathrm{N,d,P}}(i^*_{1,\mathrm{d}} - i_{1,\mathrm{d}}) \\
\frac{dd_{\mathrm{N,d,i}}}{dt} = k_{\mathrm{N,d,i}}(i^*_{1,\mathrm{d}} - i_{1,\mathrm{d}}) \\
d_{\mathrm{N,q}} = d_{\mathrm{N,q,p}} + d_{\mathrm{N,q,i}} \\
d_{\mathrm{N,q,p}} = k_{\mathrm{N,q,P}}(i^*_{1,\mathrm{q}} - i_{1,\mathrm{q}}) \\
\frac{dd_{\mathrm{N,q,i}}}{dt} = k_{\mathrm{N,q,i}}(i^*_{1,\mathrm{q}} - i_{1,\mathrm{q}}) \\
i^*_{1,\mathrm{d}} = i^*_{1,\mathrm{d,p}} + i^*_{1,\mathrm{d,i}} \\
i^*_{1,\mathrm{d,p}} = k_{\mathrm{z,d,p}}(v^*_{\mathrm{d}} - v_{\mathrm{d}}) \\
\frac{di^*_{1,\mathrm{d,i}}}{dt} = k_{\mathrm{z,d,i}}(v^*_{\mathrm{d}} - v_{\mathrm{d}}) \\
i^*_{1,\mathrm{q}} = i^*_{1,\mathrm{q,p}} + i^*_{1,\mathrm{q,i}} \\
i^*_{1,\mathrm{q,p}} = k_{\mathrm{z,q,p}}(v^*_{\mathrm{q}} - v_{\mathrm{q}}) \\
\frac{di^*_{1,\mathrm{q,i}}}{dt} = k_{\mathrm{z,q,i}}(v^*_{\mathrm{q}} - v_{\mathrm{q}}) \\
v^*_{\mathrm{d}} = v^*_{\mathrm{d,p}} + v^*_{\mathrm{d,i}} \\
v^*_{\mathrm{d,p}} = k_{\mathrm{w,d,p}}(i^*_{\mathrm{g,d}} - i_{\mathrm{g,d}}) \\
\frac{dv^*_{\mathrm{d,i}}}{dt} = k_{\mathrm{w,d,i}}(i^*_{\mathrm{g,d}} - i_{\mathrm{g,d}}) \\
v^*_{\mathrm{q}} = v^*_{\mathrm{q,p}} + v^*_{\mathrm{q,i}} \\
v^*_{\mathrm{q,p}} = k_{\mathrm{w,q,p}}(i^*_{\mathrm{g,q}} - i_{\mathrm{g,q}}) \\
\frac{dv^*_{\mathrm{q,i}}}{dt} = k_{\mathrm{w,q,i}}(i^*_{\mathrm{g,q}} - i_{\mathrm{g,q}})
\end{cases}
\tag{A2}
$$

**Table A1.** Variable definitions.

| Variable | Definition Description | Variable | Definition Description |
|----------|------------------------|----------|------------------------|
| $R$ | Machine side inductance parasitic resistance | $i^*_{1,\mathrm{q}}$ | Given q-axis inner loop current |
| $L$ | Machine side inductance | $i^*_{1,\mathrm{d,p}}$ | Given proportional component of the inner loop current of the d-axis |
| $C$ | Filter capacitor | $i^*_{1,\mathrm{d,i}}$ | Given integral component of the d-axis inner loop current |
| $R_{\mathrm{g}}$ | Grid-side inductance parasitic resistance | $k_{\mathrm{z,d,p}}$ | Proportional coefficient of the PI regulator in the d-axis |
| $L_{\mathrm{g}}$ | Grid-side inductance | $k_{\mathrm{z,d,i}}$ | Integral coefficient of PI regulator in d-axis |
| $i_{1,\mathrm{d}}$ | d-axis machine-side inductance current | $v^*_{\mathrm{d}}$ | Given d-axis middle loop voltage |
| $i_{1,\mathrm{q}}$ | q-axis machine-side inductance current | $i^*_{1,\mathrm{q,p}}$ | Given proportional component of the q-axis inner loop current |
| $v_{\mathrm{d}}$ | d-axis capacitance-voltage | $i^*_{1,\mathrm{q,i}}$ | Given integral component of the q-axis inner loop current |
| $v_{\mathrm{q}}$ | q-axis capacitance-voltage | $k_{\mathrm{z,q,p}}$ | PI proportional coefficient of control loop in q axis |
| $i_{\mathrm{g,d}}$ | inductor current of grid side in d-axis grid side | $k_{\mathrm{z,q,i}}$ | Integral coefficient of the PI regulator in the q-axis middle loop |
| $i_{\mathrm{g,q}}$ | q-axis grid-side inductance current | $v^*_{\mathrm{q}}$ | Given voltage in the q-axis middle loop |
| $d_{\mathrm{N,d}}$ | d-axis duty cycle | $v^*_{\mathrm{d,p}}$ | Given proportional component of voltage in the d-axis middle loop |
| $d_{\mathrm{N,d,p}}$ | d-axis duty cycle proportional component | $v^*_{\mathrm{d,i}}$ | Given integral component of voltage in the d-axis middle loop |

**Table A1.** *Cont.*

| | | | |
|---|---|---|---|
| $d_{N,d,i}$ | d-axis duty cycle integral component | $k_{w,d,p}$ | outer control loop PI regulator proportional coefficient in d-axis |
| $k_{N,d,P}$ | Inner control loop PI regulator proportional coefficient in axis | $k_{w,d,i}$ | Integral coefficient of PI regulator of outer control loop in d axis |
| $k_{N,d,i}$ | Integral coefficient of inner loop PI regulator in d axis | $i^*_{g,d}$ | Given d-axis outer loop current |
| $i^*_{1,d}$ | Given d-axis inner loop current | $v^*_{q,p}$ | Given proportional component of the q-axis outer loop voltage |
| $d_{N,q}$ | q-axis duty cycle | $v^*_{q,i}$ | Given integral component of the q-axis outer loop voltage |
| $d_{N,q,p}$ | q-axis duty cycle proportional component | $k_{w,q,p}$ | Proportional coefficient of outer loop PI regulator in q-axis |
| $d_{N,q,i}$ | q-axis duty cycle integral component | $k_{w,q,i}$ | Integral coefficient of outer loop PI regulator in q axis |
| $k_{N,q,P}$ | Proportional coefficient of inner loop PI regulator in q-axis | $i^*_{g,q}$ | Given q-axis outer loop the current |
| $k_{N,q,i}$ | Integral coefficient of inner loop PI regulator in q-axis | | |

$$A = \begin{vmatrix} A_1 & A_2 \\ A_3 & A_4 \end{vmatrix} \tag{A3}$$

where

$$A_1 = \begin{vmatrix} -\left(\frac{R}{L} - \frac{1}{L}\bullet k_{N,d,P}\bullet\frac{u_{d,c}}{2}\right) & -\omega & \frac{1}{L} + \frac{1}{L}\bullet k_{N,d,P}\bullet k_{z,d,p}\bullet\frac{u_{d,c}}{2} & 0 & \frac{1}{L}\bullet k_{N,d,P}\bullet k_{z,d,p}\bullet k_{w,d,p}\bullet\frac{u_{d,c}}{2} & 0 \\ \omega & -\left(\frac{R}{L} - \frac{1}{L}\bullet k_{N,q,P}\bullet\frac{u_{d,c}}{2}\right) & 0 & \frac{1}{L} + \frac{1}{L}\bullet k_{N,q,P}\bullet k_{z,q,p}\bullet\frac{u_{d,c}}{2} & 0 & \frac{1}{L}\bullet k_{N,q,P}\bullet k_{z,q,p}\bullet k_{w,q,p}\bullet\frac{u_{d,c}}{2} \\ -\frac{1}{C} & 0 & 0 & \omega & \frac{1}{C} & 0 \\ 0 & -\frac{1}{C} & -\omega & 0 & 0 & \frac{1}{C} \\ 0 & 0 & 0 & 0 & -\frac{R_g}{L_g} & -\omega \\ 0 & 0 & 0 & -\frac{1}{L_g} & \omega & -\frac{R_g}{L_g} \end{vmatrix}$$

$$A_2 = \begin{vmatrix} -\frac{1}{L}\bullet\frac{u_{d,c}}{2} & 0 & -\frac{1}{L}\bullet k_{N,d,P}\bullet\frac{u_{d,c}}{2} & 0 & -\frac{1}{L}\bullet k_{N,d,P}\bullet k_{z,d,p}\bullet\frac{u_{d,c}}{2} & 0 \\ 0 & -\frac{1}{L}\bullet\frac{u_{d,c}}{2} & 0 & -\frac{1}{L}\bullet k_{N,q,P}\bullet\frac{u_{d,c}}{2} & 0 & -\frac{1}{L}\bullet k_{N,q,P}\bullet k_{z,q,p}\bullet\frac{u_{d,c}}{2} \\ 0 & 0 & 0 & 0 & 0 & 0 \\ 0 & 0 & 0 & 0 & 0 & 0 \\ 0 & 0 & 0 & 0 & 0 & 0 \\ 0 & 0 & 0 & 0 & 0 & 0 \end{vmatrix}$$

$$A_3 = \begin{vmatrix} -k_{N,d,i} & 0 & -k_{N,d,i}\bullet k_{z,d,p} & 0 & -k_{N,d,i}\bullet k_{z,d,p}\bullet k_{w,d,p} & 0 \\ 0 & -k_{N,q,i} & 0 & -k_{N,q,i}\bullet k_{z,q,p} & 0 & -k_{N,q,i}\bullet k_{z,q,p}\bullet k_{w,q,p} \\ 0 & 0 & -k_{z,d,i} & 0 & -k_{z,d,i}\bullet k_{w,d,p} & 0 \\ 0 & 0 & 0 & -k_{z,q,i} & 0 & -k_{z,q,i}\bullet k_{w,q,p} \\ 0 & 0 & 0 & 0 & -k_{w,d,i} & 0 \\ 0 & 0 & 0 & 0 & 0 & -k_{w,q,i} \end{vmatrix}$$

$$A_4 = \begin{vmatrix} 0 & 0 & k_{N,d,i} & 0 & k_{N,d,i} \bullet k_{z,d,p} & 0 \\ 0 & 0 & 0 & k_{N,q,i} & 0 & k_{N,q,i} \bullet k_{z,q,p} \\ 0 & 0 & 0 & 0 & k_{z,d,i} & 0 \\ 0 & 0 & 0 & 0 & 0 & k_{z,q,i} \\ 0 & 0 & 0 & 0 & 0 & 0 \\ 0 & 0 & 0 & 0 & 0 & 0 \end{vmatrix}$$

**Table A2.** Eigenvalues of the state matrix.

| Eigenvalue Variable | Eigenvalue Value |
|---|---|
| $\lambda_1$ | $-5.99 \times 10^6$ |
| $\lambda_2$ | $-6.00 \times 10^6$ |
| $\lambda_3$ | $-8.42 \times 10^2 + 5.75 \times 10^3 i$ |
| $\lambda_4$ | $-8.42 \times 10^2 + 5.75 \times 10^3 i$ |
| $\lambda_5$ | $-1.24 \times 10^3 + 5.6085 \times 10^3 i$ |
| $\lambda_6$ | $-1.24 \times 10^3 - 5.6085 \times 10^3 i$ |
| $\lambda_7$ | $-7.38 \times 10^2$ |
| $\lambda_8$ | $-3.34 \times 10^2$ |
| $\lambda_9$ | $-3.13 \times 10^3$ |
| $\lambda_{10}$ | $-3.13 \times 10^3$ |
| $\lambda_{11}$ | $-1.96 \times 10^2 + 4.49 \times 10^2 i$ |
| $\lambda_{12}$ | $-1.96 \times 10^2 - 4.49 \times 10^2 i$ |

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
