# Peer review of "Seamless Switching Control Technology for the Grid-Connected Converter in Micro-Grids"

_electronics, doi:10.3390/electronics9122109_

Round 1
Reviewer 1 Report
The paper is interesting and the topic is good. I have only some minor revisions.
- No reference in the introduction to tests performed in the world
- Correct European compliant resistor symbols - Figure 6. Schematic diagram of the mode switching of the grid-connected converter mode.; Figure 7. The diagram of the single-phase grid connection;
- No discussion of errors during measurements in conclusions
Reviewer 2 Report
This paper presents seamless switching control technology for the grid-connected converter in micro-grids. The topic of this paper is interesting. The authors' work is appreciated. But there are still some aspects that can be improved to enhance the quality of the paper. Authors are asked to make some major corrections:
The Abstract in its current from is an alternative Introduction, it should clearly describe the scope with more focusing on the proposed approach and results of the study.
The three-loop control strategy for grid-connected converter is not novel. A better literature review should be provided. The authors are invited to update the section according to literature review: (i) Nonlinear Voltage Control for Three-Phase DC-AC Converters in Hybrid Systems: An Application of the PI-PBC Method, Electronics 9 (5), 847
The paper needs not only positive statements of the claimed superiority of the proposed method, but a clear statement of anything that might be lost, with thoughtful reflection on the practical difference of the predicted behaviour. The best academic papers are quite different from pure promotional material in that they should be quite open about weaknesses as well as strengths in the original content of any proposed method.
The conclusions are not summarized very good. The authors just list the main work, but the "conclusions" are not given.